# 1,4-Naphthoquinone (CNN1) Induces Apoptosis through DNA Damage and Promotes Upregulation of *H2AFX* in Leukemia Multidrug Resistant Cell Line

**DOI:** 10.3390/ijms23158105

**Published:** 2022-07-23

**Authors:** Adrhyann Jullyanne de Sousa Portilho, Emerson Lucena da Silva, Emanuel Cintra Austregésilo Bezerra, Carinne Borges de Souza Moraes Rego Gomes, Vitor Ferreira, Maria Elisabete Amaral de Moraes, David Rodrigues da Rocha, Rommel Mário Rodriguez Burbano, Caroline Aquino Moreira-Nunes, Raquel Carvalho Montenegro

**Affiliations:** 1Pharmacogenetics Laboratory, Drug Research and Development Center (NPDM), Federal University of Ceará, Fortaleza 60430-275, CE, Brazil; adrhyannportilho@gmail.com (A.J.d.S.P.); lucenaemerson@hotmail.com (E.L.d.S.); cintraemanuel@gmail.com (E.C.A.B.); betemora@ufc.br (M.E.A.d.M.); 2Institute of Chemistry, Federal University Fluminense, São João Batista St., 188-Niterói, Rio de Janeiro 24220-900, RJ, Brazil; carinneborges@id.uff.br (C.B.d.S.M.R.G.); vitorferreira@id.uff.br (V.F.); davidrocha@id.uff.br (D.R.d.R.); 3Department of Biological Sciences, Oncology Research Center, Federal University of Pará, Belém 66073-005, PA, Brazil; rommel@ufpa.br

**Keywords:** chronic myeloid leukemia, multidrug resistance, 1,4-naphthoquinone, DNA damage, genotoxicity, apoptosis

## Abstract

The multidrug resistance (MDR) phenotype is one of the major obstacles in the treatment of chronic myeloid leukemia (CML) in advantage stages such as blast crisis. In this scenario, more patients develop resistance mechanisms during the course of the disease, making tyrosine kinase inhibitors (TKIs) target therapies ineffective. Therefore, the aim of the study was to examine the pharmacological role of CNN1, a *para*-naphthoquinone, in a leukemia multidrug resistant cell line. First, the in vitro cytotoxic activity of Imatinib Mesylate (IM) in K-562 and FEPS cell lines was evaluated. Subsequently, membrane integrity and mitochondrial membrane potential assays were performed to assess the cytotoxic effects of CNN1 in K-562 and FEPS cell lines, followed by cell cycle, alkaline comet assay and annexin V-*Alexa Fluor*^®^ *488*/propidium iodide assays (Annexin/PI) using flow cytometry. RT-qPCR was used to evaluate the *H2AFX* gene expression. The results demonstrate that CNN1 was able to induce apoptosis, cell membrane rupture and mitochondrial membrane depolarization in leukemia cell lines. In addition, CNN1 also induced genotoxic effects and caused DNA fragmentation, cell cycle arrest at the G2/M phase in leukemia cells. No genotoxicity was observed on peripheral blood mononuclear cells (PBMC). Additionally, CNN1 increased mRNA levels of *H2AFX*. Therefore, CNN1 presented anticancer properties against leukemia multidrug resistant cell line being a potential anticancer agent for the treatment of resistant CML.

## 1. Introduction

Since the advent and introduction of tyrosine kinase inhibitors (TKIs), significant improvements have occurred in the treatment of chronic myeloid leukemia (CML), increasing patient’s survival rate [1,2,3,4]. Although an enormous therapeutic improvement has been demonstrated, there are still some patients that develop treatment resistance mechanisms in the course of disease, making the use with of TKIs ineffective [5,6]. In this scenario, the multidrug resistance (MDR) phenotype is well recognized in clinical practice and continues to be one of the major obstacles in CML treatment, especially in blast crisis [7,8,9,10].

Approximately 20–40% of CML patients treated with TKIs fail due to primary refractoriness (primary resistance), where patients exhibit lack of efficacy to TKIs from initiation of therapy, while secondary resistance (acquired) is defined as the loss of response [11,12,13,14]. Moreover, the responses obtained in patients in the advanced stages of CML are low and typically short-lived [15,16,17,18].

Juglone (5-hydroxy-1,4-naphthoquinone) is a natural 1,4- naphthoquinone found in the Juglandaceae, a family of compounds known for their vast biological application [19,20,21,22,23]. Although a great number of naphthoquinones have been studied in medicinal chemistry, as they possess several pharmacologic properties, these compounds have drawn attention especially for their antitumor activity [24]. In addition, the naphthoquinones are potent inhibitors of topoisomerase, DNA repair enzymes, and can also induce DNA damage [25,26,27,28].

Previously, our research group synthesized and evaluated in silico and in vitro antitumor activities of a series of synthetic naphthoquinones [18]. Among them, 5-hydroxy-2-(4-tolylthio) naphthalene-1,4-dione (CNN1), showed good pharmacokinetic profiles and cytotoxic effect against sensitive leukemia (K-562) and multidrug resistance leukemia (FEPS) cell lines. Moreover, CNN1 suppressed DNA Topoisomerase I (*TOP1*) expression in both cell lines. Therefore, this study aimed to examine whether CNN1 is able to circumscribe the multidrug resistant and induce cell death in resistant leukemia cells.

## 2. Results

### 2.1. CNN1 Induces Cytotoxicity in K-562 and FEPS Leukemia Cell Lines

Previously, Portilho and co-workers [29] demonstrated that CNN1 has excellent cytotoxicity against different leukemia cells, after 72 h of treatment. CNN1 exhibited an inhibitory effect on cell viability against K-562-Lucena-1 with an IC_50_ value of 0.90 µM (CI 95% 0.34–1.27) and presented a strong cytotoxic activity against FEPS with an IC_50_ of 0.60 µM (CI 95% 0.48–0.80) when compared to K-562 that presented an IC_50_ value of 1.12 µM (CI 95% 0.90–1.38). On the other hand, IM showed an IC_50_ of 4.97 µM (CI 95% 3.69–5.70) to K-562-Lucena-1 and 9.66 µM (CI 95% 8.45–11.1) to FEPS, a much higher concentration when compared to CNN1 in MDR cell lines. Only to sensitive cell line, K-562, IM showed a CI50 of 0.03 µM (CI 95% 0.01–0.05) (Table 1).

### 2.2. CNN1 Induces Membrane Disruption and Mitochondrial Depolarization in K-562 and FEPS Cell Lines

Flow cytometry analysis of membrane integrity (Figure 1A) showed that CNN1 (0.1 µM) significantly induced cell membrane disruption in K-562 (*p* < 0.001) and also in FEPS cells (*p* < 0.001) after 24 h of treatment (Figure 1B,C). Additionally, IM at 0.1 µM was able to significantly disrupt the membrane integrity only in K-562 cells (*p* < 0.01).

In order to investigate whether CNN1 compound was able to cause disruption of the mitochondrial membrane potential, rhodamine 123 was used after 24 h of treatment (Figure 1D). Naphthoquinone CNN1 (0.1 µM) induced a significant (*p* < 0.001) disruption of mitochondrial membrane potential in K-562 and FEPS cell lines (Figure 1E,F), when compared to negative control. In addition, IM (0.1 µM or 10 µM) induced a significant mitochondrial membrane depolarization in K-562 (*p* < 0.001) and FEPS (*p* < 0.001) cell lines. 

### 2.3. CNN1 Induces DNA Fragmentation, Cell Cycle Arrest and DNA Damage in Leukemia Cell Lines

Cell cycle analysis was measured to evaluate whether the CNN1 compound induces effects in the cell cycle of K-562 and FEPS cell lines. The negative control presented 32.21% of cells in G0/G1 phase, 30.30% in S phase and 25.67 in G2/M phase. We found that CNN1 (0.1 µM), after 24 h of treatment, triggered a significant accumulation of 39.9% of cells in G2/M phase (*p* < 0.001) and induced a significant decrease in cells in the S phase (18.27%) in K-562 cells (*p* < 0.05) (Figure 2A). On FEPS cell line, CNN1 (0.1 µM) also caused cell cycle arrest of 40.70% at G2/M phase (*p* < 0.001) and increased the number of cells at G0/G1 phase to 25.66% (*p* < 0.05) (Figure 2B). Treatment with IM (0.1 µM or 10 µM) decrease the percentage of cells at G0/G1 phase in K-562 (*p* < 0.01) and FEPS (*p* < 0.05) cells. In addition, negative control presented 7.46% of K-562 cell at sub-G1 and 5.84% of FEPS at sub-G1, while CNN1 treatment increases the number of cells at sub-G1 to 10.85% and 9.35% (*p* < 0.01), respectively (Figure 2C,D). The IM also induces increases in sub-G1 in K-562 (*p* < 0.01) and FEPS (*p* < 0.001) cells.

The comet assay was performed to determine genotoxic potential of CNN1 (0.1 µM) and IM (0.1 µM) in K-562 and FEPS leukemia cell lines and PBMC after 3 h of exposure. CNN1 induced DNA damage in leukemia cell lines. CNN1 (0.1 µM) significantly induced DNA damage of K-562 and FEPS cells in comparison to negative control (*p* < 0.001). Statistical analysis showed that CNN1 was more genotoxic to leukemia cells than to non-malignant peripheral mononuclear blood cells (PBMC). Moreover, IM showed significant differences between leukemia cells and PBMC, where IM present significant genotoxic effects only in K-562 (Figure 2E,F).

### 2.4. CNN1 Induces Apoptosis in K-562 and FEPS Cell Lines

In order to investigate whether CNN1 compound promotes apoptosis after 24 h of treatment, annexin V-*Alexa Fluor*^®^ *488* and PI were used by flow cytometry (Figure 3A). After exposure of CNN1 (0.1 µM), the results showed a significant increase (*p* < 0.001) in early apoptotic cells (Annexin-V positive/PI negative) and an increase in late apoptosis (Annexin V-positive/PI-positive) in K-562 and FEPS cell lines (*p* < 0.001) (Figure 3B,C), when compared to the negative control, demonstrating the apoptosis as a cell death pathway. IM (0.1 µM or 10 µM) also showed a significant increase in apoptotic cells in leukemia cells. 

### 2.5. CNN1 Significantly Increased H2AFX Gene Expression in K-562 and FEPS Cell Lines

We also investigated whether CNN1 induces *H2AFX* gene modulation after 18h of treatment. H2AFX is an important biomarker to monitor genotoxic events [29]. Samples treated with CNN1 (0.1 µM) showed a significant increase in *H2AFX* gene expression (*p* < 0.001) in K-562 and FEPS cell lines when compared to control, as show in Figure 4. Furthermore, we also determined whether CNN1 altered *ABCB1* mRNA expression, and no significant difference was found in *ABCB1* expression after 18 h of drug exposure (*p* > 0.06) in K-562 and FEPS cell lines. The results also showed no significant differences in *BCR-ABL1* expression in both K-562 (*p* > 0.27) and FEPS (*p* > 0.28) cell lines. 

## 3. Discussion

The naphthoquinones present different pharmacological properties, among them anticancer action, antifungal, antibacterial and antiviral activities, as well as allelopathic activities [30,31,32,33]. There are many studies that describe the effect of naphthoquinones, or its analogs, in several type of tumors and demonstrate an in vitro and in vivo chemotherapeutic potential of these compounds [34,35,36]. However, this is the first study that identified the antitumor effect of naphthoquinone against resistant leukemia cells (FEPS). The development of *MDR* is still a significant obstacle to provide effective treatment to many patients with cancer, including CML. In this sense, one strategy to overcome MDR is the use of compounds able to selectively target MDR cells [37].

Our previous study showed that CNN1 is more potent to resistance cell lines FEPS (IC50 0.60 µM) and K-562-Lucena-1 (IC50 0.90 µM) when compared to a parental sensitive cell line K-562 (IC50 1.12 µM) [28]. Moreover, it was observed that CNN1 is less toxic to normal human cells such as fibroblasts (IC50 15.34 µM) when compared to leukemia cell lines. Our results suggest that CNN1 might cause few side effects; however, clinical studies are still needed to confirm this hypothesis. In vitro experiments conducted by other authors demonstrate that 1,4-naphthoquinones also presented a less toxic effect in peripheral blood mononuclear cells (PBMC) [26]. Although CNN1 showed low toxicity to normal human cells, other research described that naphthoquinones are toxic to human lymphocytes [38]. 

To demonstrate the efficacy of CNN1 on resistant (FEPS) and sensitive (K-562) leukemia cell lines, experiments were performed to evaluate if CNN1 was able to induce apoptosis as a cell death mechanism. Analyzing the cell membrane integrity after treatment with CNN1, a significant difference was found between the leukemic cell lines when compared to positive control (*p* < 0.001). In fact, CNN1 caused pronounced rupture in membrane integrity on leukemia cells compared to IM standard therapy. Some naphthoquinones can induce disruption of membrane integrity [39]. Our presented data corroborates a study by Montenegro and co-workers [40] that also described that 1,4-naphthoquinone induced alteration of cell membrane integrity (instability) on an HL-60 cell line, a model derived from acute myeloid leukemia. Another study showed that HL-60 cells treated with naphthoquinone (β-lapachone) also presented disruption of membrane integrity, but only in low concentrations [41]. These data reinforce the results presented in this paper and suggest that CNN1 is a compound with important anticancer properties. 

The next question was to determine whether disruption of membrane integrity would be linked with cell death by depolarization of mitochondrial membrane potential, considered one of the hallmarks of apoptosis [42]. In the current study, CNN1 caused depolarization of mitochondrial membrane potential in both leukemia cell lines and was more pronounced when compared to IM (*p* < 0.001). Previous studies have reported a decrease in mitochondrial membrane potential induced by 1,4-naphthoquinone in liver cancer cells (Hep3B), leukemia cells (HL-60), and human breast adenocarcinoma cells (MCF-7) [26,43,44]. The deregulation of apoptotic pathways induces leukemic cells to accumulate, thus promoting further genetic alterations and also the leukemogenesis, therefore, mitochondrial damage may be a promising strategy in the regulation of apoptosis in CML [45]. The mechanism of apoptosis mainly consists of two core pathways that are the extrinsic pathways initiated by activation of membrane-bound death receptors leading to caspase cleavage [46] and the intrinsic pathway is a mitochondrial-mediated pathway [47]. These findings support the involvement of intrinsic apoptosis pathways in cell death induced by CNN1 (*p* < 0.001), mediated by the loss of mitochondrial membrane potential.

It was also investigated whether the pronounced effect of CNN1 on leukemia multidrug resistant cell line could reflect a cell cycle arrest. Our results showed that CNN1 induced cell cycle arrest at G2/M phase for both K-562 *(p* < 0.001) and FEPS *(p* < 0.001) cell lines. Tumor cells present an increased rate of cellular proliferation, and this feature of uncontrolled cell division can be targeted to treat cancer patients [48]. The primary anticancer therapies that include chemotherapy and ionizing radiation induce cell death caused by DNA damage. Thus, the DNA molecule is one of the main targets of these agents because DNA replication is essential on cell cycle progression [49]. Therefore, genotoxic compounds are attractive candidates to improve chemotherapy [50]. In agreement with our results, previous data observed that other naphthoquinones also induced arrest at G2/M phase in cancer cell lines with a prominent MDR phenotype, including lung cancer [51] and breast cancer [52]. 

Furthermore, CNN1 showed a possible preferential genotoxic effect against K-562 and FEPS cells and did not cause significant DNA damage in peripheral blood mononuclear cells (PBMC). Data from previous work by our group showed that CNN1 reduced *TOP1* expression levels in both leukemia cell lines [28]. Thus, this evidence suggests that suppression of *TOP1* expression can be correlated with a genomic damage induced by CNN1 [53]. 

Some naphthoquinones have exhibited topoisomerase inhibition effects by catalytic activity or trapping of *TOP1,* that can result in DNA damage [54]. These data consolidate the selective genotoxic effect of CNN1 to MDR phenotype in the FEPS cell line and suggest that CNN1 can be considered a promising lead compound with reduced side effects for development of new chemotherapeutic regimens [55].

In this research, the apoptosis induced by CNN1 was confirmed by phosphatidylserine externalization in K-562 (*p* < 0.001) and FEPS (*p* < 0.001) cell lines, similar findings were reported on in vitro models of breast cancer [56], lung adenocarcinoma [57], and colon carcinoma [33]. Apoptosis deregulation is a remarkable hallmark of cancer, enabling cancer cells to develop MDR phenotype, additionally, this feature had been highlighted as the major cause of chemotherapy failure in CML [58]. The externalization of phosphatidylserine is a well-known apoptosis phenotypic characteristic, caused by an increase in mitochondrial permeability and release of cytochrome c and its accumulation in cytosol [59]. Our results showed that CNN1-mediated cytotoxicity is based on its ability to induce apoptosis by DNA fragmentation, loss of mitochondrial membrane potential and phosphatidylserine externalization.

In addition to the mechanism of cell death induced by CNN1 in K-562 and FEPS cell lines mediated by DNA damage, we investigated the H2AFX biomarker [29]. Our results indicate the hypothesis that CNN1 induces overexpression of *H2AFX* in K-562 (*p* < 0.01) and FEPS (*p* < 0.01) cells, suggesting that this upregulation of *H2AFX* gene has a major role in initiating cell death in leukemia MDR cells through the DNA damage signaling pathway. To the best of our knowledge, our study is first to describe *H2AFX* gene overexpression in mediated apoptosis of resistant leukemia cells (FEPS). 

Several studies have shown that deregulation of the DNA damage response (DDR) pathway causes genomic instability in non-neoplastic cells and thus compromising tumor cell sensitivity to anticancer therapy [60,61,62]. H2AFX plays a key role in DNA damage response and is important to DNA repair proteins signaling, mainly at sites that chromatin suffered a possible damaged, and for activation of checkpoint proteins, that blocks the cell cycle progression [63,64,65]. 

Other authors also described that *H2AFX* low expression compromise the apoptotic response to IM and promote blast crisis of CML, as well as the *H2AFX* deficiency, promotes B-cell tumorigenesis, suggesting that deregulation could make cells more sensitive to leukemogenic factors [64]. In particular, the role of *H2AFX* as a tumor suppressor is involved to its regulation of apoptosis [63]. Zangh and co-workers [65] reinforce that *H2AFX* dysfunction is considered one important event in cancer patients that develop resistance during treatment, including IM therapy, and demonstrated the importance of H2AFX in the apoptosis regulation process of CML cells. 

At last, we evaluated whether CNN1 could decrease the mRNA expression of *P-glycoprotein* (*P-gp*), encoded by *ABCB1* (ATP–binding cassette, subfamily B, member 1), and the expression of *BCR-ABL1* transcripts. In our results, CNN1 did not cause decreased expression of *BCR-ABL1 and ABCB1* in both K-562 and FEPS cell lines. This result corroborates a previous in silico study by our group, where CNN1 was not considered inhibitor or substrate of P-gp [28]. In addition, Sales and co-workers [66] described that FEPS cells presents a low transcript expression level of *BCR-ABL1* compared to K562, indicating that the *BCR–ABL1*-independent mechanism of resistance, that occurs in CML patients with overexpression of *P-glycoprotein* (*P-gp*) is also found in the FEPS cell line model [67]. This finding reinforces that P-gp overexpression is the main mechanism of resistance found in FEPS cell line, and this in vitro model mimics the molecular events of MDR and underlines the poor adherence to IM, that is observed in non-responder patients [68]. Furthermore, Eaddie and co-workers [69] showed that patients that are refractory or intolerant to IM treatment presented high levels of *ABCB1* mRNA, and they also failed to respond to subsequent treatment with nilotinib and dasatinib. Finally, recent studies demonstrated some evidence that IM, nilotinib, and dasatinib are Pgp-substrates [70,71]. In this scenario, our finding suggests that CNN1 induces apoptosis in resistant leukemia cells by inducing DNA damage throughout *H2AFX* modulation. Moreover, CNN1 can lead to new drugs that might be capable of overcoming the interference of efflux transporters in MDR phenotype. 

## 4. Materials and Methods

### 4.1. Ethics Aspects and Lymphocytes Isolation

Peripheral blood mononuclear cells (PBMC) were obtained from healthy donors, non-smoking individuals (two female and two males, ages 20–30 years), after a signed written consent form to participate in the study which was approved by the Ethics Research Committee of the Federal University of Ceará (registration number 52352121.0.0000.5054). Blood samples were collected according to Fenech (2012) [72]. The lymphocytes were isolated using Histopaque-1077 (Sigma-Aldrich, St. Louis, MO, USA) for comet assay. Briefly, 1 × 10^6^ lymphocytes were seeded into six-well plates in Roswell Park Memorial Institute medium 1640 (RPMI 1640, Gibco^®^, New York, NY, USA), supplemented with 10% fetal bovine serum (FBS, Gibco^®^), 100 U/mL of penicillin, 100 μg/mL of streptomycin (Gibco^®^) and phytohemagglutinin A (PHA; Gibco-Invitrogen, Waltham, MA, USA) for 20 h at 37 °C.

### 4.2. Cell Culture

Chronic myeloid leukemia cell lines (K-562), vincristine-resistant derivative (K-562-Lucena1), daunorubicin-resistant derivative (FEPS). K-562, K-562-Lucena1 and FEPS was provided by Prof. Dr. Vivian Rumjanek from the Federal University of Rio de Janeiro [73]. The cells were cultured in Roswell Park Memorial Institute medium 1640 (RPMI 1640, Gibco^®^). The cell lines were supplemented with 10% fetal bovine serum (FBS, Gibco^®^), 100 U/mL of penicillin, and 100 μg/mL of streptomycin (Gibco^®^). All cells were cultured at 37 °C and 5% CO_2_.

### 4.3. CNN1 and Chemicals 

Stock solution of CNN1 (1mM) was dissolved in DMSO (0.02%) from Sigma^®^, and Imatinib Mesylate (IM) (1mM) were used as a positive controls. In other studies conducted by our group, CNN1 demonstrated cytotoxic activity against K-562, K-562-Lucena1 and FEPS cells after 72 h. The concentration of CNN1 (0.1 µM) used in tests was based in CI50 determined before [28].

### 4.4. Leukemia Cell Lines 

K-562 sensitive cell line, derived from a patient with CML. The K-562-Lucena-1 cell line is derived from the K-562 cell line following selection by vincristine and presenting *ABCB1* overexpression [68], and the FEPS cell line, also derived from K-562 following selection by daunorubicin and presenting *ABCB1* and *ABCC1* overexpression [74]. In the case of K-562-Lucena-1 and FEPS, the medium was maintained with vincristine sulfate at a final concentration of 60 nM and daunorubicin at a concentration of 466 nM, respectively. The following experiments were performed only in K-562 and FEPS cell lines based in previous study [28]. 

### 4.5. Cell Viability Assay

The cytotoxicity of the CNN1 and IM were evaluated in a concentration-response *curve* against leukemia cell lines (K-562 and FEPS). Cells were plated (5 × 10^3^ cells/well) in a 96-well-plate, after 24 h cells were treated with compounds (0.312–20 µM) for 72 h. DMSO (0.02%) was used as a negative control. Cell viability was evaluated by the Alamar blue assay [75]. Briefly, Alamar blue solution (0.2 mg/mL) was added to each well for 3 h at 37 °C. The fluorescence was read at 535 nm (reduced form; resorufin) and 595 nm (oxidized form; resazurin) using a microplate reader (Beckman Coulter Microplate Reader DTX 880, Bio-TEK).

### 4.6. Membrane Integrity by Propidium Iodide (PI)

Cells (K-562 and FEPS) were seeded into 6-well plates at a density of 50 × 10³ per well and maintained for 24 h at 37 °C in incubator. After this period, cells were submitted to treatment with CNN1 (0.1 µM) for 24 h. IM (0.1 µM) was used as a positive control and DMSO (0.02%) as negative control. Posteriorly, cells were suspended in PBS and incubated in propidium iodide (1 μg/mL) solution for 30 min at 37 °C, in the dark. Fluorescence intensity (from 10.000 cells) was immediately analyzed by flow cytometry (BD FACSVerse) with 488-nm laser excitation [76].

### 4.7. Mitochondrial Membrane Potential Analysis

K-562 and FEPS cell lines (50 × 10^3^ cells per well) were plated into 6-well for 24 h at 37 °C. Cells were then treated with CNN1 at concentration 0.1 µM for 24 h. IM (0.1 µM or 10 µM) was used as a positive control and DMSO (0.02%) as a negative control. Next, cell pellets were resuspended in rhodamine 123 solution (1 μg/mL) for 20 min at 37 °C, in the dark. Then, mitochondrial transmembrane potential (ΔΨm) was determined using flow cytometry system (BD FACSVerse) [77]. 

### 4.8. Cell Cycle Analysis

Cells were plated at a density of 50 × 10^3^ cells per well into 6-well plates for 24 h at 37 ˚C. After 24 h, cells were treated with CNN1 at 0.1 µM for 24 h. DMSO (0.02%) as negative control. IM (0.1 µM) was used as a positive control. Cells were pelleted and fixed with 70% alcohol for 30 min at 4 °C. After, cells were submitted to Triton- X100 (0.1% v/v in PBS). Subsequently, cells are incubated for 40 min at 37 °C and propidium iodide was added (50 µg/mL). Events (10.000) were evaluated by flow cytometry (BD FACSverseTM). All data were analyzed using FlowJo Software v.10 [78].

### 4.9. Alkaline Comet Assay

K-562, FEPS and PBMC were plated into 6-well plates at (5 × 10^3^) per well for 24 h at 37 °C in a CO_2_ incubator. Then, cells were exposed to CNN1 at concentration (0.1 µM) for 3 h. IM (0.1 µM) was used as positive control and DMSO (0.02%) as negative control. Following exposure, the cells were suspended in 150 μL in 0.8% of low melting agarose (LMPA) in phosphate buffered saline (PBS) without calcium and rapidly spread on three slides pre-coated with 1.5% normal melting agarose (NMA) and waited to solidify at 4 °C for 5 min. After, slides containing cells were placed in the chilled lysis solution containing 100 mM EDTA, 2.5 M NaCl, 10 mM Tris, 10% DMSO and 1% Triton X-100, pH 10) at 4 °C for 24 h. The slides were incubated is fresh electrophoresis buffer (300 mM NaOH and 1 mM EDTA, pH ≥ 13.0) for 20 min in other to unwinding the DNA. Then, horizontal electrophoresis at 300 mA and 34 V for 20 min was performed. Finally, slides were washed with distilled water for 5 min and fixed with absolute ethanol. Thereafter, slides were stained with propidium iodide at 20 µg/mL A total of 100 comets were counted for each sample using a fluorescence microscope (Kinetic Imaging, Liverpool, UK) [79].

### 4.10. Analysis of Apoptosis

K-562 and FEPS cell lines were seed into 12-well plates at a 50 × 10³ per well for 24 h. After, cells were exposed to CNN1 (0.1 µM) for 24 h. The positive control used IM (0.1 µM) and DMSO (0.02%) was used as negative control. Subsequently, cells were washed with PBS and resuspended in 200 µL Annexin-binding buffer consisting of 4 μL Annexin V-FITC and then incubated with propidium iodide for 20 min. After incubation, cells were resuspended in 1× binding buffer. The fluorescent emission was measured by flow cytometry system (BD FACSVerse) [80].

### 4.11. RNA Isolation

K-562 and FEPS cell lines (50 × 10^3^ cells per well) were plated into 6-well plates for 24 h at 37 °C. After 24 h, cells were treated with CNN1 (0.1 μM) for 18 h, a time that does not modify cellular viability (Appendix A). After treatment, cells were collected for mRNA extraction using TRIzol Reagent^®^ (Invitrogen™) according to the manufacturer’s protocol. The RNA concentration and quality were determined by using Nanodrop 2000 (ThermoFisher Scientific, Waltham, MA, USA). From 20 ng of RNA, the cDNA was synthesized using HighCapacity cDNA Reverse Transcriptase kit (Life Technologies, Carlsbad, CA, USA) to convert the extracted and purified RNA to cDNA. The conversion step was performed in a Veriti^®^ thermal cycler (Applied Biosystems^®^, Foster City, CA, USA) [81].

### 4.12. mRNA Expression Analysis

All requirements proposed in the Minimum Information for Publication of Quantitative Real-Time PCR Experiments–MIQE Guidelines were followed [82]. The experiments were performed using Fast SyberGreen kit (Applied Biosystems^®^, Foster City, CA, USA) for H2A.X Variant Histone (*H2AFX)* gene and Beta-Actin (*ACTB)* gene as an endogenous control. The other genes selected for expression evaluation were *ABCB1* (Hs00184500_m1), *BCR-ABL1* (Hs03024541_ft) and the *ABL* gene (Hs99999002_mH) as the endogenous control, and each sample used the concentrations following: 3 μL of cDNA, 1 μL of each primer/probe, 12.5 μL of TaqMan^®^ Gene Expression Master Mix or Fast SyberGreen kit (Life Technologies’), and 8.5 μI of Ultrapure water. The RT-qPCR was performed using QuantStudio5 Real-Time PCR system (Applied Biosystems^®^, Foster City, CA, USA). Each sample was performed in triplicate for the validation of the technique and the values of CT and gene expression levels were calculated using the 2^−ΔΔCT^ (delta-delta threshold cycle) [83] using DMSO control as a calibrator of the experiment (Appendix A).

### 4.13. Statistical Analyses

Each assay was performed in triplicate from three independent experiments. All data were expressed as mean ± standard deviation (SD), the distribution of normality was verified by the Kolmogorov–Smirnov test and statistically compared to untreated control (DMSO) by Analysis of Variance (ANOVA) followed by Bonferroni’s considering a (*p* < 0.05), and was performed by GraphPad Prism 5.0 software. 

## 5. Conclusions

CNN1 naphthoquinone presented cytotoxicity activity based on its ability to induce apoptosis in K-562 and FEPS cell lines. Briefly, we showed that CNN1 blocks cell cycle progression, inducing DNA damage followed by the recruitment of *H2AFX* gene signaling, that seems to play a major role in initiating of apoptosis by intrinsic pathway. Therefore, these results indicated that CNN1 has a promising anticancer activity and as a potential lead compound to the development of a new treatment for CML patients who fail their TKI therapy due to intolerance and/or developed resistance, mainly in refractory disease advantage stages.

## Figures and Tables

**Figure 1 ijms-23-08105-f001:**
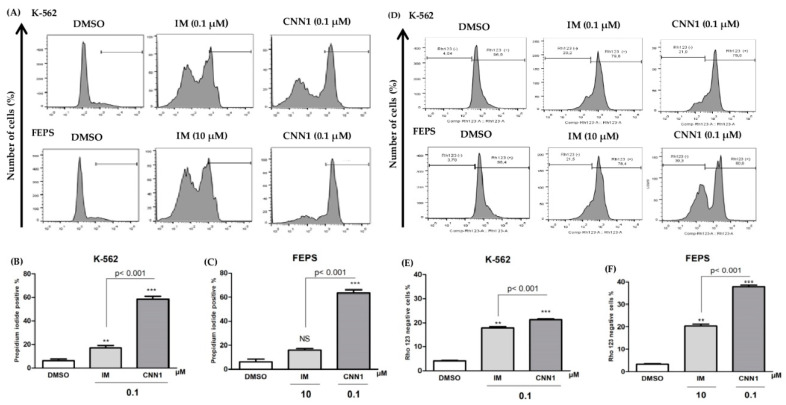
CNN1 caused a significant rupture in the membrane integrity and induced mitochondrial depolarization on K-562 and FEPS cells, after 24 h treatment (0.1 µM): (**A**) Detection of PI of the negative control (DMSO) and treated groups; (**B**) CNN1 significantly affected the membrane integrity in K-562 cells; (**C**) CNN1 can also induce disruption in membrane integrity in FEPS cells; (**D**) identification of Rhodamine 123 in negative control (DMSO) and treated groups; (**E**) Rhodamine 123 negative cells after treatment with CNN1 in K-562 cells; (**F**) CNN1 induces mitochondrial membrane depolarization in FEPS after CNN1 exposure. Data from membrane integrity are presented as mean ± SD of three independent experiments. Treated samples were compared to DMSO and statistically analyzed by ANOVA followed by Bonferroni’s post-test. Significant differences: ** *p* < 0.01, *** *p* < 0.001. NS = non-significant.

**Figure 2 ijms-23-08105-f002:**
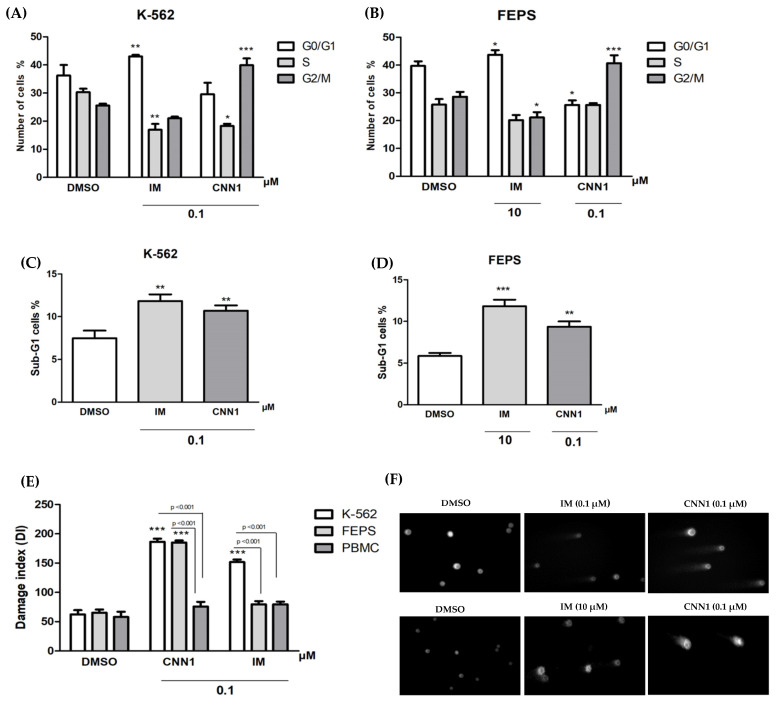
Induction of DNA fragmentation, cell cycle arrest and genotoxic effect after treatment with CNN1 in leukemia cells and PBMC by alkaline comet assay. IM was positive control: (**A**) Effect of CNN1 in cell cycle in K-562 cell line; (**B**) activity of CNN1 in cell cycle in FEPS cell line; (**C**) representation of effect of CNN1at number of cells in sub-G1 of K-562; (**D**) number of cells in sub-G1 phase of K-562 and FEPS cell lines after CNN1 exposure; (**E**) alkaline comet assay in leukemia cell lines and PBMC; (**F**) distribution of DNA migrations from negative control (DMSO) and treated groups, using light microscope. The number of cells in G0/G1, S and G2/M phase was calculated using *FlowJo*™ software. Treated samples were compared to DMSO. The bars represent the mean ± standard error of the mean of three independent experiments. Significant differences compared to control (DMSO) * *p* < 0.05, ** *p* < 0.01, *** *p* < 0.001 by ANOVA followed by Bonferroni post-test.

**Figure 3 ijms-23-08105-f003:**
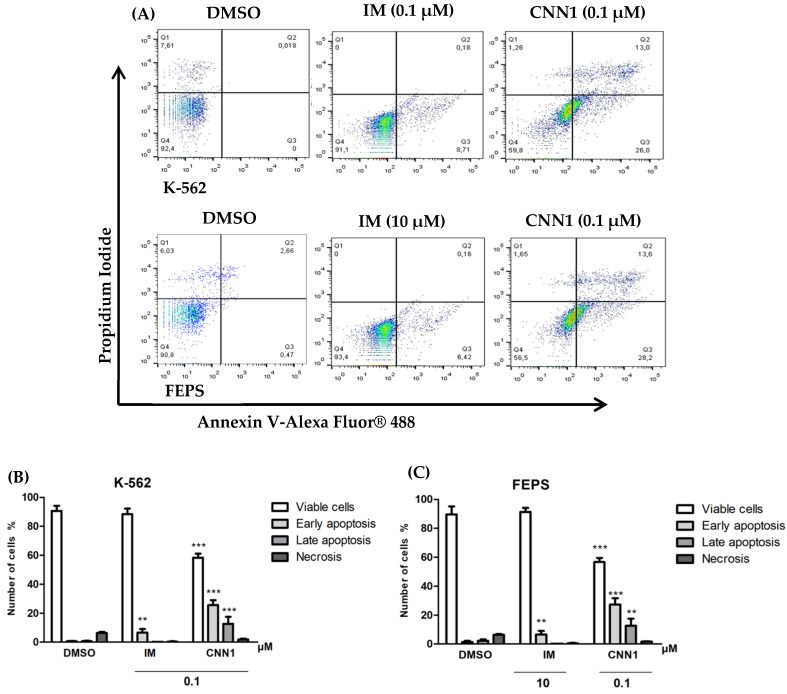
CNN1 increased apoptosis in K-5622 and FEPS cell lines. Percentage of apoptotic cells after 24 h treatment with CNN (0.1 µM); cells were labeled with annexin-V and PI and the result was analyzed by flow cytometry: (**A**) Representation of Dot Plot graph of the negative control (DMSO) and treated groups; (**B**) effect of CNN1 on cell death in K-562 cell line; (**C**) CNN1 induced cell death in FEPS cell line. Bars represents the percentage of viable cells, early apoptosis, late apoptosis, and necrosis. Results are expressed as mean ± SD of three independent experiments. Treated samples were compared to DMSO. Significant differences compared to control (DMSO) ** *p* < 0.01, *** *p* < 0.001 by ANOVA followed by Bonferroni post-test. Q1, necrosis; Q2, late apoptosis; Q3, early apoptosis and Q4 viable cells.

**Figure 4 ijms-23-08105-f004:**
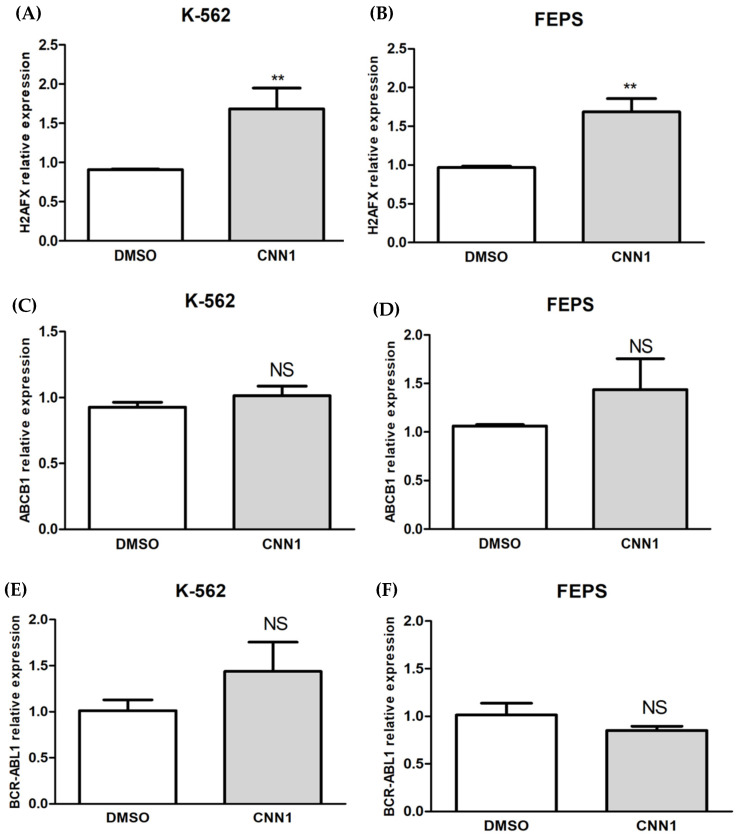
Relative gene expressions of *H2AFX*, *ABCB1* and *BCR-ABL1* in K-562 and FEPS cell lines after 18h of treatment: (**A**) CNN1 significantly modulated *H2AFX* expression in K-562 cell line; (**B**) CNN1 also demonstrated an increased *H2AFX* expression in FEPS cell line; (**C**) no significant *ABCB1* expression was observed after treatment with CNN1 in K-562 cell line; (**D**) CNN1 did not caused significant differences in *ABCB1* expression after treatment in FEPS cell line; (**E**,**F**) CNN1 did not significantly modulated *BCR-ABL1* expression in K-562 and FEPS cell lines, respectively. Results are expressed as the mean ± SD of three independent experiments. Treated samples were compared to DMSO and statistically analyzed by t-test. Significant differences: ** *p* < 0.01. NS = non-significant.

**Table 1 ijms-23-08105-t001:** Cytotoxic activity expressed as IC_50_ in µM and with its respective confidence interval of 95% in K-562, K-562-Lucena-1 and FEPS cell lines, after 72 h of exposure.

IC_50_ (µM *^a^*)			
Compounds	K-562	K-562-Lucena-1	FEPS
**CNN1**	1.12	0.90	0.60
	(0.90–1.38)	(0.34–1.27)	(0.48–0.80)
***^b^* IM**	0.03	4.97	9.66
	(0.01–0.05)	(3.69–5.70)	(8.45–11.1)

*^a^* Data are presented as IC_50_ values and 95% confidence intervals obtained by nonlinear regression analysis of three independent experiments. *^b^* Imatinib Mesylate (IM) was used as positive control.

## Data Availability

All individuals consented to the acknowledgement.

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
