# Peer review of "1,4-Naphthoquinone (CNN1) Induces Apoptosis through DNA Damage and Promotes Upregulation of H2AFX in Leukemia Multidrug Resistant Cell Line"

_ijms, 2022, doi:10.3390/ijms23158105_

Round 1

Reviewer 1 Report

In the manuscript by Portilho et al. the important issue of developing new drugs to overcome the multi-drug resistance in leukemia has been arisen. Generally, the manuscript is clear and logical, the performed experiments are appropriate. I have just couple of minor comments:

1. In the methods parapraph 2.5 the authors used DMSO 0.1% instead of 0.2% as in other experiments and solution stock. Is there a reason for the difference?

2. In paragraph 2.11 in Methods should be stated which plates have been used, as the authors write cells/well and not cells/cm2.

3. Though small but multiple English mistakes have to be carefully proven (e.g. lines 45, 49, 90-92, 102,  152, 213, 216, 351, 405, 407 etc). Please, also note, that K562 is not an erythroleukemia, but just a cell line model (line 96). Of special stylistic attention is the line 201, as I hope that the authors do not develop "a candidate against patients", but rather against the disease.

Congratulations to the authors for their successful effort.

Author Response

Dear reviewer, my co-authors and I would like to thank you for the suggestions made during this high-quality review and then we present the answers to the questions.

We inform that with the reviews and suggestions, we were able to improve the idea presented by our work and we appreciate the opportunity. We hope this review has left the article suitable for publication in this high-impact journal and respect in the area.

Kind Regards.

Response to reviewer 1

In the manuscript by Portilho et al. the important issue of developing new drugs to overcome the multi-drug resistance in leukemia has been arisen. Generally, the manuscript is clear and logical, the performed experiments are appropriate. I have just couple of minor comments:

  1. In the methods parapraph 2.5 the authors used DMSO 0.1% instead of 0.2% as in other experiments and solution stock. Is there a reason for the difference?

R = The reviewer is correct, and we apologize for this mistake, the concentration of DMSO (0.1%) was replaced for the correct heading to DMSO (0.2%) in the methods paragraph 2.5.

  1. In paragraph 2.11 in Methods should be stated which plates have been used, as the authors write cells/well and not cells/cm2.

R= We appreciate the observation, and the corrected information was added in the manuscript.

  1. Though small but multiple English mistakes have to be carefully proven (e.g. lines 45, 49, 90-92, 102, 152, 213, 216, 351, 405, 407 etc).

R= We appreciate the comments, and the multiple English mistakes were corrected.

Please, also note, that K562 is not an erythroleukemia, but just a cell line model (line 96). Of special stylistic attention is the line 201, as I hope that the authors do not develop "a candidate against patients", but rather against the disease.

R= We appreciate the observation and K-562 was describe in the manuscript as cell line derived from a patient with CML.  In addition, we corrected the paragraph and add that our “results showed that CNN1 as a potential lead compound for refractory or intolerant TKIs patients in advantage CML stages”.

Reviewer 2 Report

The authors presented work on the cytotoxic action of the CNN1 naphthoquinone derivative mainly on K562 and FEPS cell lines. Through various methodological approaches, the data shown in this manuscript support that CNN1 causes cell death by apoptosis in treated cells, potentially triggered by the alteration of mitochondrial function. The authors also affirm a genotoxic action of CNN1, but the data presented do not fully support this hypothesis.

Comments are given below.

In the abstract, the authors seem to suggest that experiments have been replicated on PBMC cells. On the contrary, only the comet assay was performed on PBMC cells. The abstract must be properly corrected.

Line 210, it is unclear what the authors mean by membrane disruption, is there a known direct effect of CNN1 on membrane integrity?

Figure 1A seems not to be the correct one: The authors indicate the integrity of the membrane using PI, while the panel shows a pattern of RHO123

Indicate how PI positive and RHO123 negative cells were calculated

Report cytofluorimeter plots showing cell cycle data.

Report the comet assay figures.

The modulation of TOP1 expression does not necessarily imply an alteration of the TOP1 enzyme activity. Similarly, the alteration of the TOP1 activity has been reported for some naphthoquinones, but no data on the effect of CNN1 on TOP1 activity nor a direct interaction of CNN1 with DNA has been presented in this manuscript. Considering these premises, the title is misleading and needs to be changed, and the discussion needs to be properly corrected.

Author Response

Dear reviewer, my co-authors and I would like to thank you for the suggestions made during this high-quality review and then we present the answers to the questions.

We inform that with the reviews and suggestions, we were able to improve the idea presented by our work and we appreciate the opportunity. We hope this review has left the article suitable for publication in this high-impact journal and respect in the area.

Kind Regards.

Response to reviewer 2

The authors presented work on the cytotoxic action of the CNN1 naphthoquinone derivative mainly on K562 and FEPS cell lines. Through various methodological approaches, the data shown in this manuscript support that CNN1 causes cell death by apoptosis in treated cells, potentially triggered by the alteration of mitochondrial function. The authors also affirm a genotoxic action of CNN1, but the data presented do not fully support this hypothesis.

Comments are given below.

In the abstract, the authors seem to suggest that experiments have been replicated on PBMC cells. On the contrary, only the comet assay was performed on PBMC cells. The abstract must be properly corrected.

R= The reviewer is correct, and we appreciate the comments. We perform the correction in the abstract, and exhibit the results of PBMC cells only in comet assay.

Line 210, it is unclear what the authors mean by membrane disruption, is there a known direct effect of CNN1 on membrane integrity?

R= In this study, we investigate the mechanism of cell death by CNN1 and our results showed that naphthoquinone CNN1 induced significant disruption of membrane integrity on K-562 and FEPS cell lines by flow cytometry. In fact, several research demonstrated that naphthoquinones could cause alteration of cell membrane integrity (MONTENEGRO et al. 2010; GOLEVA et al. 2020). Thus, the mechanism of cell death by CNN1 has direct effect on membrane integrity of leukemia cell lines.

Figure 1A seems not to be the correct one: The authors indicate the integrity of the membrane using PI, while the panel shows a pattern of RHO123. Indicate how PI positive and RHO123 negative cells were calculated.

R= The reviewer is correct, and we apologize for this mistake. The figure of RHO123 was replaced for the correct PI positive.

Report cytofluorimeter plots showing cell cycle data.

R= We appreciate the comments and agree with the referee, however, due to covid-19 pandemic we were not able to extract of histogram of cell cycle on time. In this context, we exhibit only the graphic.  

Report the comet assay figures.

R= We appreciate the observation. The figures of alkaline comet assay were added to the negative control (DMSO) and treated groups using the light microscope.

The modulation of TOP1 expression does not necessarily imply an alteration of the TOP1 enzyme activity. Similarly, the alteration of the TOP1 activity has been reported for some naphthoquinones, but no data on the effect of CNN1 on TOP1 activity nor a direct interaction of CNN1 with DNA has been presented in this manuscript. Considering these premises, the title is misleading and needs to be changed, and the discussion needs to be properly corrected.

R= We thank the comments. Previously, our research group observed that CNN1 induces TOP1 gene expression modulation in both cell lines by RT-qPCR (Portilho et al. 2021). Thus, our reasoning is that suppression of TOP1 can caused DNA replication stress and result in the DNA damage, confirmed by our alkaline comet assay, and this information was added in the manuscript. Therefore, these results reinforce the possible alteration of the TOP1 function. Although we considering the necessity of confirmed the effect of CNN1 on TOP1 activity.

In addition, the reviewer is correct when mentioned that our data didn’t demonstrate a direct interaction of CNN1 with DNA, since we only presented in this manuscript the DNA damage by CNN1 using alkaline comet assay. Therefore, the title and discussion were also altered to be properly.

Round 2

Reviewer 2 Report

The authors provided a new version of the manuscript, in which the main concerns raised in the first revision have been addressed. The present version is suitable for publication on IJMS.